# Pixel-Based Soil Loss Estimation and Prioritization of North-Western Himalayan Catchment Based on Revised Universal Soil Loss Equation (RUSLE)

Shishant Gupta [1,*], Chandra Shekhar Prasad Ojha [1], Vijay P. Singh [2], Adebayo J. Adeloye [3] and Sanjay K. Jain [4]

[1] Department of Civil Engineering, Indian Institute of Technology, Roorkee 247667, India; cspojha@gmail.com
[2] Department of Biological & Agricultural Engineering, and Department of Civil & Environmental Engineering, Texas A&M University, College Station, TX 77843, USA; vijay.singh@ag.tamu.edu
[3] Institute of Infrastructure and Environment, Heriot-Watt University, Edinburgh EH14 4AS, UK; a.j.adeloye@hw.ac.uk
[4] Water Resources Systems Division, National Institute of Hydrology, Roorkee 247667, India; sjain.nihr@gov.in
[*] Correspondence: sgupta2@ce.iitr.ac.in

**Abstract:** Land degradation is a noteworthy environmental risk causing water quality issues, reservoir siltation, and loss of valuable arable lands, all of which negate sustainable development. Analysis of the effect of land use changes on erosion rate and sediment yield is particularly useful to identify critical areas and define catchment-area treatment plans. This study utilized remote sensing and geographical information system/science (GIS) techniques combined with the Revised Universal Soil Loss Equation (RUSLE) on a pixel basis to estimate soil loss over space and time and prioritized areas for action. The methodology was applied to the Sutlej catchment from the perspective of sedimentation of the Bhakra reservoir, which is leading to the loss of active storage capacity and performance and of the safety and efficiency of many existing hydroelectric projects in the Sutlej and its tributaries that drain the Himalayas. Soil loss estimation using RUSLE was first calibrated using data from three sites, and the calibrated model was then used to estimate catchment soil loss for 21 years (1995–2015). The number of land use/land cover (LULC) classes as 14 and the C factor as 0.63 for agriculture land were optimized using the observed data for the Sutlej catchment. Further, the linkage between soil erosivity and annual precipitation was also established. It was concluded that extensive control treatment would be necessary from the soil and water conservation point of view. Structures like check dams, terraces, bunds, and diversion drains in the upstream can overcome the issue of fragmentation of soil in the Sutlej catchment.

**Keywords:** land degradation; soil erosion; RUSLE; Himalayas; remote sensing; GIS





## 1. Introduction

Land degradation due to the erosion of soil affects agricultural productivity due to the detachment of nutrients from the topsoil. This also leads to increased sediment flux into reservoirs, reducing the active storage capacity [1,2]. Approximately 45% of the land in the Himalayas and the Western Ghats in India needs soil conservation planning and measures because of soil loss due to erosion [3]. It is estimated that about three million hectares of land is eroded annually into the northeastern Himalayas due to shifting cultivation and prevailing high slopes [4]. Soil erosion from the entire Himalayan region depends on the basin characteristics, climate, basin topography, and land-use pattern, but anthropogenic activities also play a major role [5,6]. Factors such as deforestation, construction of roads on a large scale, mining, and agriculture on higher slopes all increase soil erosion and sediment concentration in the higher and lower Himalayan rivers and contaminate the entire region. These variables also increase the chances of larger floods, which carry extremely high sediment loads [7,8]. It has been observed that soil and water conservation measures and

land management practices are useful for reducing the peak rate of runoff and soil loss in the catchment [1–3,9–12].

There are several methods for the estimation of soil loss and sediment yield from different landforms [4]. Remote sensing using a GIS platform delivers an extensive amount of temporal data that aids in predicting sedimentation and soil loss on a pixel-by-pixel basis and prioritizing erosion-prone areas, while conventional processes for doing so are time-consuming, repetitive, and costly [1,5,13–16]. It is challenging to assess or forecast soil loss on a catchment or regional scale because of the varying terrain, small number of gauging sites, and land use factors in the mountainous area [17,18]. Advancements in remote sensing and geospatial technology using GIS make it a resourceful platform for observing, analyzing, mapping, and the board of regular assets, such as sediment yield and soil disintegration, on a substantial scale [9,19]. Basic observational models are still utilized for soil disintegration and residue yield expectations for their effortlessness, which make them pertinent regardless of whether just a constrained measure of input information is accessible. Basic strategies, for example, the Universal Soil Loss Equation (USLE) after Wischmeier and Smith [20,21] or the Revised Universal Soil Loss Equation (RUSLE) after Renard [22], are often utilized for the estimation of soil erosion from catchments [3,19,23–26]. The RUSLE equation has been broadly utilized as a prescient model in the assessment of soil loss and the impacts of various management practices for over 40 years. It utilizes the same technique as USLE does, in addition to a new equation for slant length and steepness and new preservation practice standards that enhance the accuracy of elements of the USLE model [17,18,27]. Many researchers [1,2,7,11,28–33] indicate that advancements in RS and GIS techniques are an asset in the characterization and prioritization of catchments with the spatial outline of soil destruction present in the catchment on a large scale [34,35]. The temporal variation of sediment yield and, in particular, erosion rate have been studied by several researchers [9,19]. However, the effect of land use/land cover (LULC) changes on both erosion rate and sediment yield has not been studied. In this research work, the impact of land use/land cover (LULC) changes on erosion rate and sediment yield was analyzed simultaneously in a mountainous region.

The aim of this analysis was to (1) assess the impact and spatial pattern of soil loss on a pixel-by-pixel basis for the Sutlej basin for 21 years using RUSLE; (2) validate the approach to sediment yield estimation using available observed data of sediment concentration at different locations over time; (3) examine the influence of land use/land cover variation on soil loss in the entire Sutlej catchment; and (4) prioritize sub-catchments on the basis of erosion rate.

## 2. Study Area and Datasets

The Sutlej basin area considered for the study is about 21,595 km$^2$ and located between 75°39′5.06″ and 79°38′51.45″ E longitude and 30°50′56.22″ and 32°59′56.94″ N latitude, having an elevation range of 359 m to 6750 m above MSL (mean sea level) (Figure 1). The Sutlej River is perennial in nature because a large area of the catchment contributes to snowmelt, and it is estimated that more than 50% of the runoff in the Sutlej River is from snowmelt. The Sutlej River travels about 322 km inside the Tibetan Domain and then enters India near Shipkilla (Tashigang village). The Sutlej River then travels about 300 km up to Bhakra Dam. Many major canals stretch out from the Sutlej River, indicating the river's importance as a waterway for agriculture and power generation. The Beas and Chenab rivers, the two largest tributaries, enter from the right. The Dipalpur, Pakpattan, Panjnad, Sirhind, and Bikaner canals are all major irrigation systems that originate from the Sutlej. The area has varying climatic and topographic conditions and is widely used for irrigation and hydropower production The large Bhakra Nangal Dam has been constructed in India at the point where the river emerges from the mountains. The Bhakra Beas Management Board (BBMB), created by the Government of India in 1966, is responsible for its management and distribution.

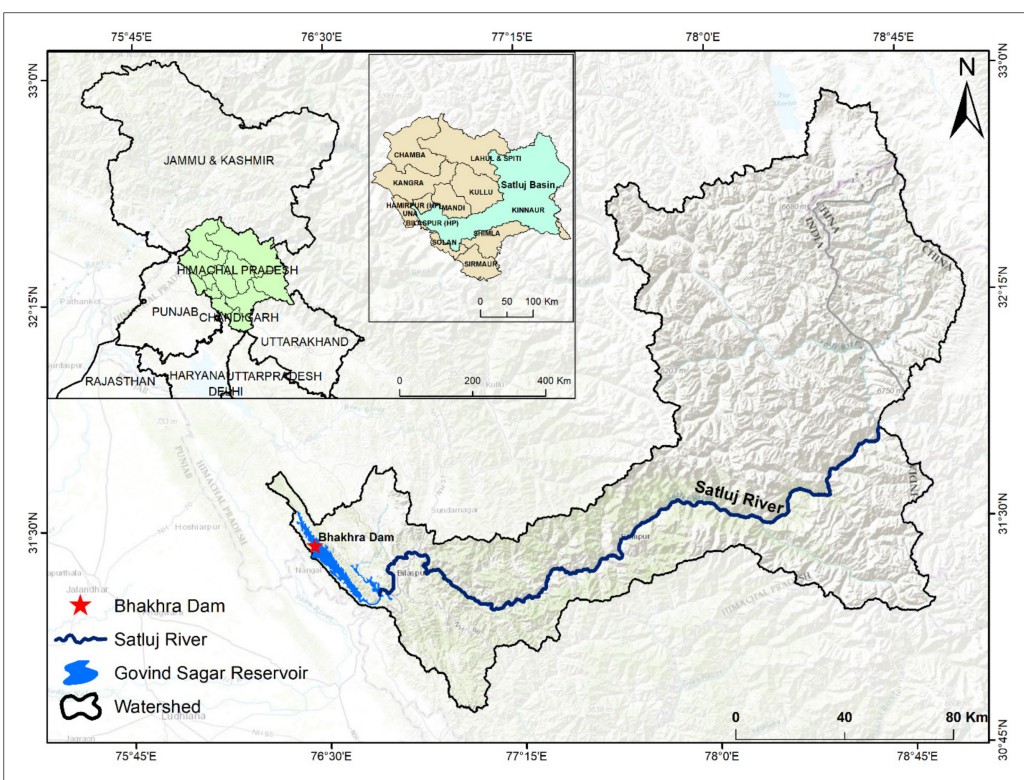

**Figure 1.** Location map of the Sutlej catchment.

### 2.1. Hydrogeology of Sutlej Basin

Sutlej catchment hydrology is governed by the South Asian monsoon and spring and summer season snowmelt in the northern Himalayas. Heavy rainfall in the catchment that causes extensive flooding in the downstream section is due to summer monsoons. In the year 1955, maximum flood discharge had been recorded when the river flowed at about 16,990 cubic meters per second. In the winter season, the flow is lesser than in the rainy and summer seasons because of the lesser amount of rainfall and melt water from the Himalayan glaciers. On the Sutlej River, the world's highest gravity dam is constructed at the downstream side of the Govind Sagar reservoir, and the Nangal barrage is constructed downstream of the Bhakra Nangal Dam. Tattapani, Namgia, Kalpa, Bilaspur, Suni and Rampur are some of the well-known human settlements that originated along the banks of Sutlej River [8]. During the monsoon season, precipitation rises together with altitude up to a certain point, beyond which it gradually decreases. The outer and middle Himalayas get most of their precipitation during the monsoon season. It is known that the yearly rainfall in the outer and central Himalayas is 46% and 41%, respectively, due to monsoon rains. When compared to other Himalayan ranges, the middle Himalayas' orography has a more significant impact on precipitation and snowfall. Snowfall seems to vary more dramatically with altitude than rainfall. Rainfall drops exponentially, and snowfall rises linearly with altitude in the broader Himalayan range. At altitudes over 4000 m on the protected side of the broader Himalayan range, precipitation becomes almost non-existent [35]. Extremely random and unsystematic precipitation distribution and dramatic topographic change characterize the Sutlej River basin. The pattern of rainfall provides evidence of such impacts [36].

### 2.2. Soil of Sutlej Basin

It is relevant to know the soil type, as it is useful for estimating erodibility. The Sutlej River catchment has nearly poor sandy loam soil, constituting uncropped substratum, and stony soil. The soil in the study area has been grouped as Udalts—Ochrepts (shallow and brown in color), Othents—Ochrepts (shallow, red, loamy, and sandy, which is suitable

for horticulture), Udoll soil (cold desert), and glaciers and snow cap soils. Medium deep, well-drained soil with loamy surface has been observed in the lower reach of the Sutlej with a limited area (Figure 2). Using Figure 2, the availability of soil types in the catchment can be seen—glaciers, sandy, loamy skeletal, glaciers and rock outcrops, loamy, and rock outcrops—according to their coverage and weightage. Table 1 shows the spatial variation of soil types and erodibility factors in that order.

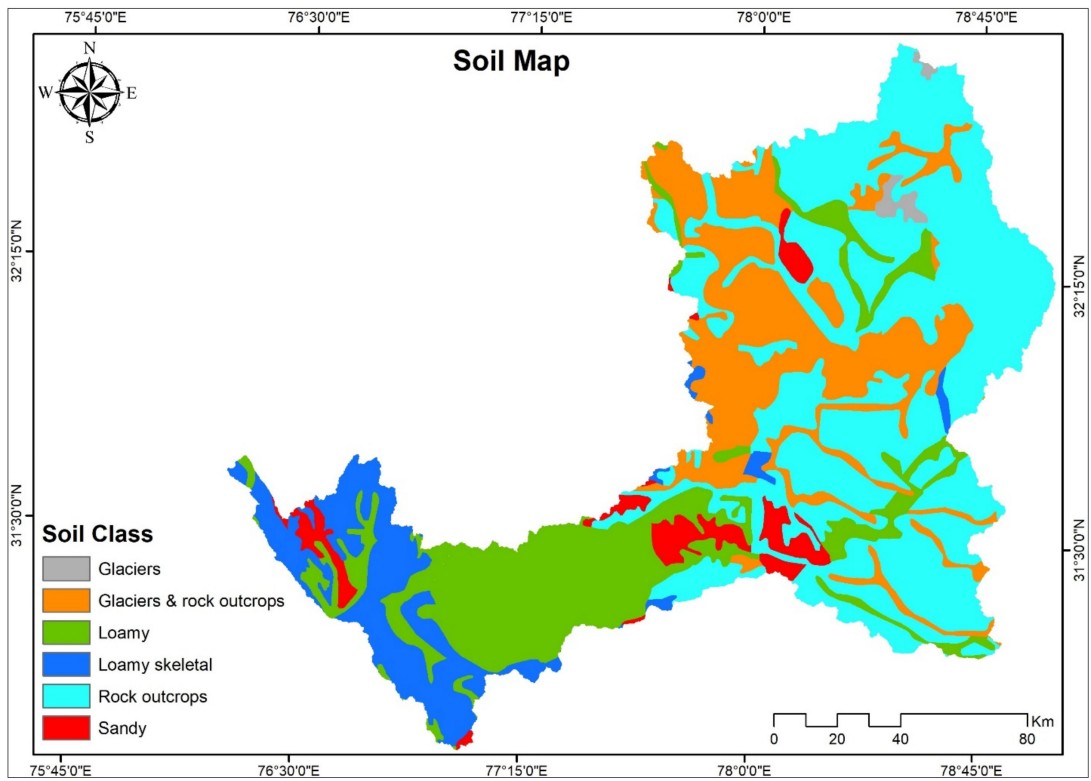

**Figure 2.** Soil map of the study area.

**Table 1.** Soil types and erodibility (K) factor in the Sutlej catchment.

| S. No. | Soil Classes | Area (km²) | Area (%) | K Factor |
|:---:|---|:---:|:---:|:---:|
| 1 | Glaciers | 141.06 | 0.65 | 0 |
| 2 | Sandy | 924.18 | 4.28 | 0.44 |
| 3 | Loamy skeletal | 2215.60 | 10.26 | 0.19 |
| 4 | Glaciers and rock outcrops | 4338.80 | 20.09 | 0.01 |
| 5 | Loamy | 4422.39 | 20.48 | 0.27 |
| 6 | Rock outcrops | 9554.31 | 44.24 | 0.02 |

### 2.3. Climate of the Sutlej Basin

The climate of the Sutlej basin varies because of elevation variations. The diversity in the climate of the Sutlej basin is seen from the hot and sub-humid tropical climate in the southern part to the glacier and the alpine climate in the eastern and northern parts of the basin. The monsoon season runs from July to September, followed by a warm October. The cold season of the year occurs from November to February, with an average temperature varying from 0 °C to 15 °C. Snowfall is common in the alpine regions. There are three major climatic zones that run north to south across the study region, each defined by the average annual precipitation and average annual temperature: semi-arid to arid zone, sub-humid to humid zone, and wet to humid zone [37,38]. Climate, geography, and other environmental elements all play a role in distinguishing one region from another.

### 2.4. Data Availability

Figure 3 and Table 2 provide relevant information about the rain gauge stations. The Thiessen polygon method and the inverse distance weighted (IDW) interpolation technique were used for computing the average rainfall over the entire catchment, because rainfall is not uniform in the catchment and the density of rain gauges is much higher in the lower region than in the middle and upper regions. Locations and Thiessen weights of rain gauge sites are shown in Figure 3, and details are presented in Table 2. Satellite images, soil map and digital elevation model, and rainfall data were utilized for the evaluation of soil loss, as depicted in Table 3.

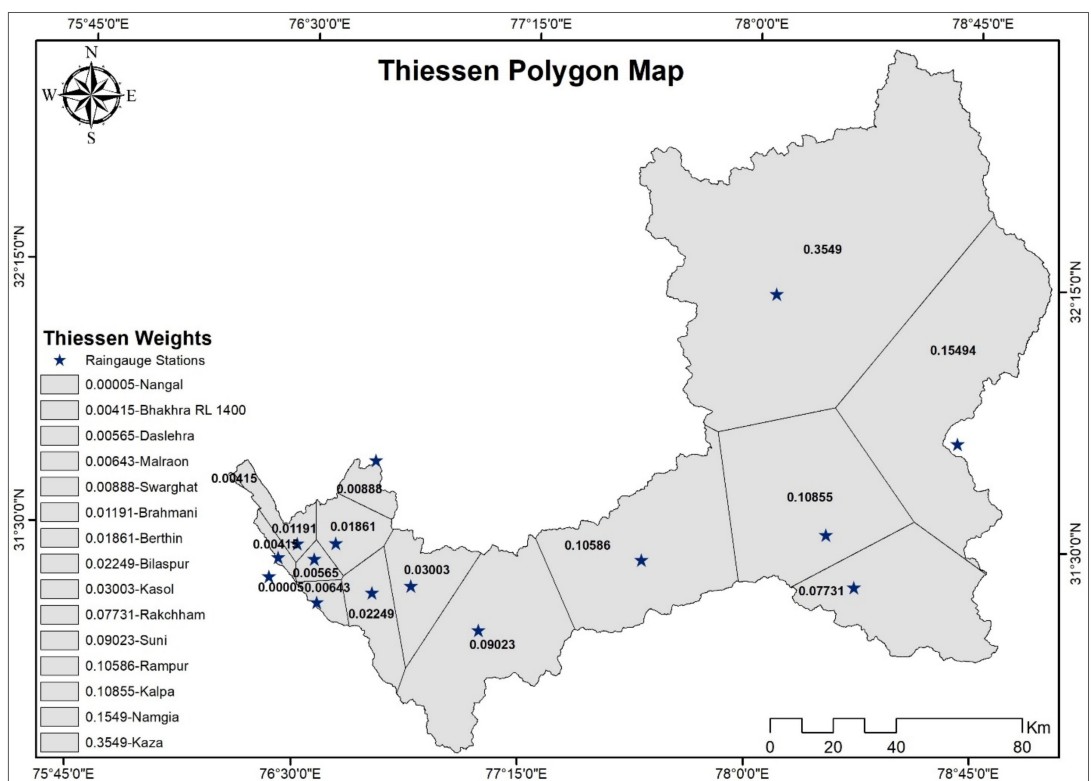

**Figure 3.** Rain gauge stations and Thiessen weight map.

**Table 2.** Details of rain gauge locations and their Thiessen weights.

| S. No. | Site Name | Longitude (E) | Latitude (N) | Area (km$^2$) | TW |
|---|---|---|---|---|---|
| 1 | Berthin | 76.622 | 31.471 | 401.94 | 0.01861 |
| 2 | Bhakhra | 76.432 | 31.424 | 89.73 | 0.00416 |
| 3 | Bilaspur | 76.750 | 31.333 | 485.58 | 0.02249 |
| 4 | Brahmani | 76.495 | 31.465 | 257.21 | 0.01191 |
| 5 | Daslehra | 76.553 | 31.423 | 122.04 | 0.00565 |
| 6 | Kalpa | 78.258 | 31.540 | 2344.13 | 0.10855 |
| 7 | Kasol | 76.878 | 31.357 | 648.43 | 0.03003 |
| 8 | Kaza | 78.072 | 32.225 | 7664.24 | 0.3549 |
| 9 | Malraon | 76.567 | 31.300 | 138.89 | 0.00643 |
| 10 | Namgia | 78.692 | 31.808 | 3346.01 | 0.15494 |
| 11 | Nangal | 76.404 | 31.368 | 1.133 | 0.00005 |
| 12 | Rakchham | 78.356 | 31.392 | 1669.53 | 0.07731 |
| 13 | Rampur | 77.644 | 31.454 | 2286.05 | 0.10586 |
| 14 | Suni | 77.108 | 31.238 | 1948.60 | 0.09023 |
| 15 | Swarghat | 76.746 | 31.713 | 191.86 | 0.00888 |

**Table 3.** Information on the datasets.

| Type of Data | Data Source | Summary |
|---|---|---|
| Digital Elevation Model (DEM) | www.usgs.gov (accessed on 30 April 2023) | ASTER DEM (30 m Spatial Resolution) |
| Satellite Image | www.usgs.gov (accessed on 30 April 2023) | Landsat-8 Image (with 30 m Spatial Resolution) |
| Soil Data | National Bureau of Soil Survey and Land Utilization Planning (NBSS and LUP), India | Soil map of 2010. Six classes were found on the basis of their texture |
| Rainfall Data | Bhakhra-Beas Management Board (BBMB) and Indian Meteorological Department (IMD), India | Rainfall data for a period of 13 years (1995–2007) provided by BBMB and for 8 years (2008–2015) provided by IMD with 15 rain gauge stations |

## 3. Methodology

### 3.1. Observed Sediment Data

Sediment sampling and data collection as well as discharge measurements were done on a daily basis by the Bhakra Beas Management Board (BBMB) at Kasol, Rampur, and Suni in the Sutlej catchment. Sediment concentration data are available in the categories of coarse sediment (above 0.2 mm), medium sediment (0.075 to 0.2 mm), and fine sediment (below 0.075 mm) in t gm/litre and discharge in cusecs. Sediment production in tonnes per hectare per year was generated using the unit conversion technique. Observed sediment concentration data at Kasol, Suni, and Rampur were available for the period of 1995 to 2005 and 1995 to 2003 for Rampur on a daily basis and were provided by the Bhakra Beas Management Board (BBMB), Chandigarh, Punjab India.

### 3.2. Estimation of Soil Loss in the Sutlej Basin

It is helpful to evaluate soil loss from a catchment on a grid basis, because it reduces the constraints on the implementation of watershed management programs and provides measures for controlling soil loss with the help of RS and GIS. It also prioritizes areas critically susceptible to erosion in the catchment [1,2,5,30]. Using Horton's law for stream area, stream length, and stream numbers, the Sutlej basin was divided into nine sub-watersheds [39,40]. Watershed delineation for the Sutlej catchment up to the Bhakra Nangal dam was created using the hydro-processing tool in ArcGIS, and the operation unit for soil loss estimation in this catchment was a 30 × 30 m grid. Higher relief and sparse vegetation are the two prime measures, as both can lead to erosion in the catchment, but it has been observed that 50% of the catchment is covered by snow and ice, and there is not much erosion in the higher region of the catchment as compared to the lesser Himalayas, which is more prone to soil loss [5]. Drainage density in the Sutlej catchment is low, and the pattern of drainage is dendritic in nature and is of a 5th order type catchment. It has been seen that the numbers of the 1st, 2nd, 3rd, 4th, and 5th order streams are 249, 58, 11, 3, and 1, respectively. The whole catchment does not produce the same amount of sediment yield and soil loss [2,8] uniformly, so it can be the criterion for prioritization of watersheds. Using the RUSLE equation [22] soil loss was calculated on a grid basis, and the required parameters for this calculation were generated in an ArcGIS environment using RS and GIS. Estimated soil loss was compared with observed data for the years 1995 to 2005 for Kasol and Suni, but for Rampur, it was compared with data for the period of 1995 to 2003 due to the unavailability of observed data of sediment concentration, and the model could not be validated for the rest of the period, i.e., 2006 to 2015. Grid-based soil loss estimation helps with the effective management plan and control of soil loss with suitable measures [41]. A flow chart of the methodology is also shown in Figure 4.

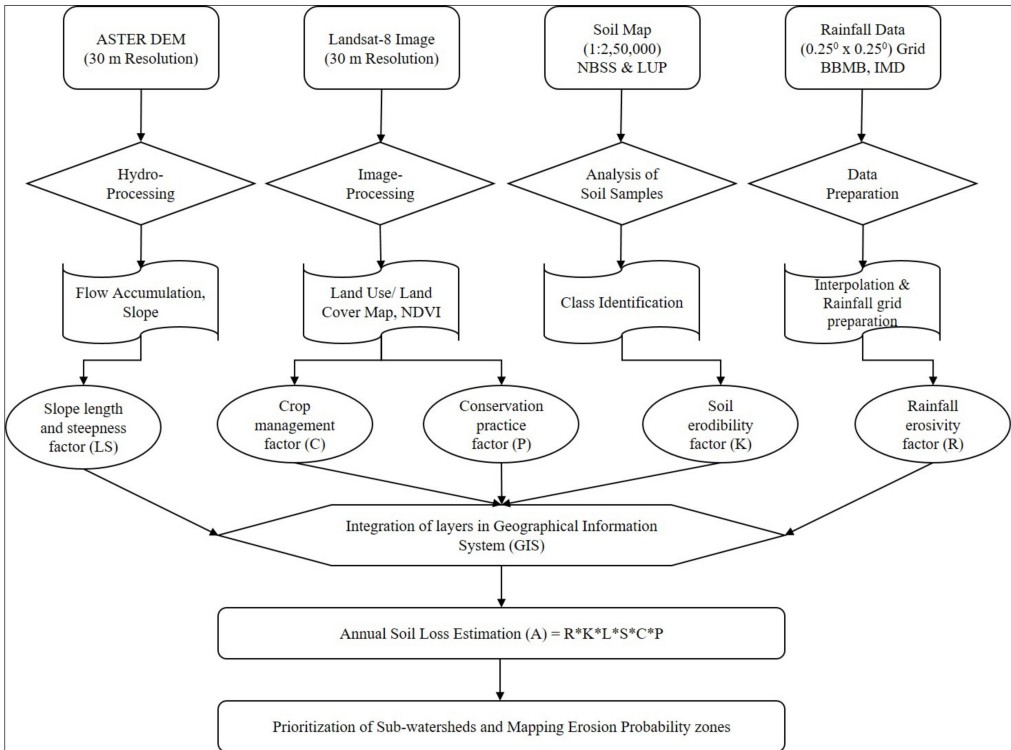

**Figure 4.** Methodology for average annual soil loss estimation in the Sutlej catchment.

### 3.3. Data Processing and RUSLE Parameter Estimation

The RUSLE scheme [22,42], an improved form of USLE [20], was utilized for the assessment of average annual soil erosion potential. The RUSLE equation requires five input variables in the raster format in GIS, such as erosivity, erodibility, slope length and steepness factor, crop management, and conservation practice factor, for annual soil loss estimation on a pixel basis and was considered to foresee longtime average soil losses in overflow from explicit field territories.

The RUSLE equation is as follows:

$$\mathbf{A} = \mathbf{R} \times \mathbf{K} \times \mathbf{L} \times \mathbf{S} \times \mathbf{C} \times \mathbf{P} \tag{1}$$

where A is the loss of soil rate per unit area, stated in the units selected for K and for the period selected for R. A is expressed in (tons/ha/year), but other units can be selected (i.e., ton/km$^2$/year); R defines the rainfall-runoff erosivity factor for a specific field plus a factor for any significant runoff from snowmelt (MJ-mm-ha$^{-1}$ h$^{-1}$ year$^{-1}$); K represents the erodibility factor—the soil loss rate per unit area of specified soil as measured on a standard plot, which is defined as a 22.1 m length of a uniform 9% slope in continuous clean-tilled fallow, expressed in (t-h-ha/ha/MJ/mm); L and S are dimensionless topographic factors that represent the slope length and steepness of a catchment, respectively; C is the crop management factor (dimensionless) for a specific field; and P is the dimensionless protection practice factor, which accounts for soil loss as a ratio of a particular practice, such as contouring or strip cropping, to the corresponding soil loss up and down the slope [21]. Various steps involved in the methodology are summarized in Figure 4.

#### 3.3.1. Rainfall Erosivity Factor (R)

Erosivity (R) is conventionally governed by computing the kinetic energy for each equivalent magnitude of a rainstorm times the amount of rainfall for the period; at that point, these are summed and increased by the extreme 30 min average annual value [21].

The following equation, developed by [20] and modified by [43], was used in the computation of rainfall erosivity surface:

$$R = 1.735 \times 10\hat{}\left(1.50 \times \log \sum_{i=1}^{12} \frac{Pi^2}{P} - 0.8188\right) \tag{2}$$

where R is the erosivity factor (MJ-mm-ha$^{-1}$ h$^{-1}$ year$^{-1}$), P$_i$ is the monthly rainfall (mm), and P is the annual rainfall (mm). The average values of erosivity ranged from 180.6 to 1518.7 MJ-mm-ha$^{-1}$ h$^{-1}$ year$^{-1}$. Figure 5 represents the erosivity map developed using precipitation data for the Sutlej catchment.

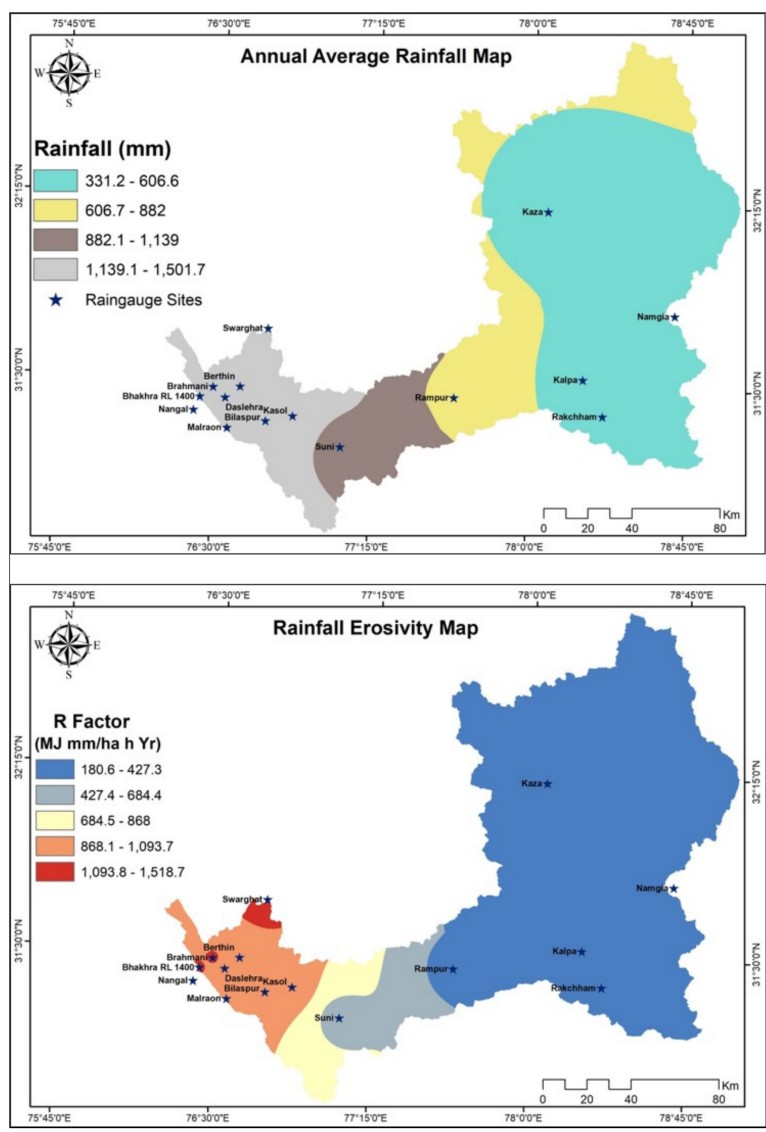

**Figure 5.** Average rainfall and erosivity (R) map of the catchment.

3.3.2. Soil Erodibility Factor (K)

Erodibility (K) can be defined as a proportion of vulnerability of soil elements to damage per unit of erosivity for an identified soil on a unit plot having a 9% uniform slope and a slope length of 22.13 m over a persistently orderly abandoned land with an all over inclination surface. K naturally ranges from about 0.10 to 0.45, with high-sand and high-clay content soils having lower standards and high-silt content soils having higher standards [22,27]. In RUSLE, erodibility is supposed to be determined consistently.

As per Wischmeier [20,21],

$$K = \frac{2.1 \times M^{1.14} \times 10^{-4} \times (12 - OM) + 3.25 \times (s - 2) + 2.5 \times (p - 3)}{100} \quad (3)$$

where K is the soil erodibility (t ha h/MJ-mm), OM is the percentage organic matter, p is the soil permeability code, s is the soil structure code, and M is a function of the primary particle size fraction given by

$$M = (\%silt + \%very\ fine\ sand) \times (100 - \%clay) \quad (4)$$

Using Equations (3) and (4), the spatial variation of erodibility values of soil in the catchment is shown in Figure 6.

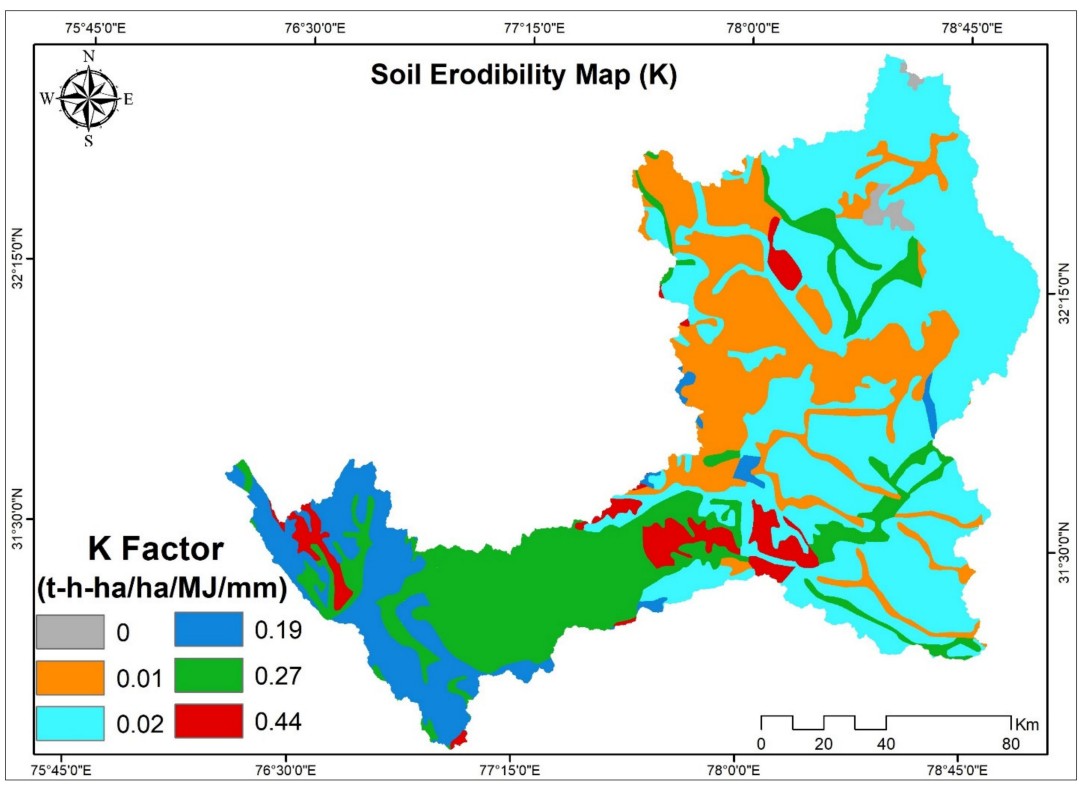

**Figure 6.** Soil erodibility (K) distribution map of the Sutlej catchment.

### 3.3.3. Slope Length and Slope Steepness Factor (LS)

Slope length (L) and steepness (S) factors play a key character in RUSLE for soil loss estimation and show the effect of topographical variation on erosion in the catchment [44]. The slope length factor (L) was calculated using the following formula given by [44,45]:

$$L = \left(\frac{\lambda}{22.13}\right)^m \quad (5)$$

where $\lambda$ is the field slope length (m), 22.13 is the RUSLE unit plot length (m), and m is a variable slope-length exponent, which is dimensionless in nature and which depends on the slope steepness, being 0.5 for slopes exceeding 5%, 0.4 for 4% slopes, and 0.3 for slopes less than 3%. The topographic factor (LS) map and its spatial distribution for the Sutlej catchment using ASTER DEM are shown in Figure 7, while a 30 m gird was used as the field slope length ($\lambda$). The assumption for slope length was the same for many researchers [2,3,33,44,46,47]. The field slope length ($\lambda$) is characterized as the horizontal separation from the beginning of runoff to the point, where either (1) the slant

angle diminishes enough that deposition starts, or (2) overflow ends up amassed in a characterized channel. Thus, soil loss varies as the field slope length varies.

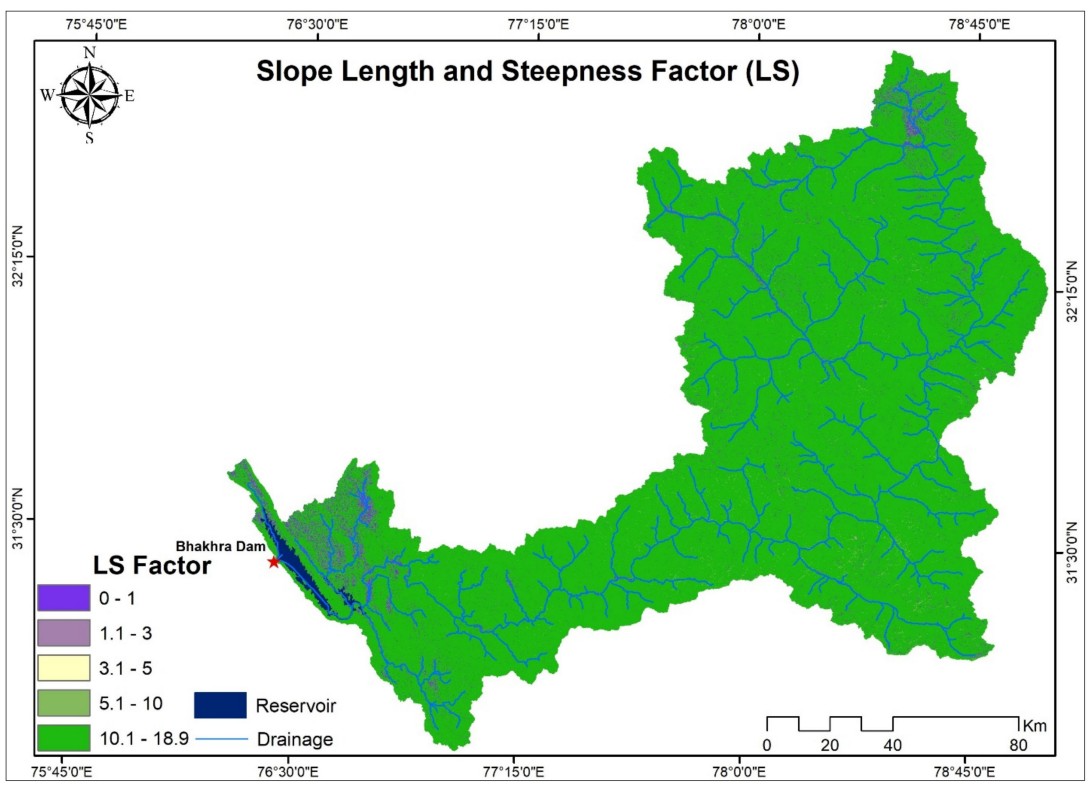

**Figure 7.** Topographic (LS) factor of the Sutlej catchment.

The slope-length exponent m was calculated as

$$m = \beta/(1+\beta) \tag{6}$$

$$\beta = (\sin\theta/0.0896)/[3.0 \times (\sin\theta)^{0.8} + 0.56] \tag{7}$$

where θ is the slope angle.

The slope steepness factor (S) was evaluated [44,45]:

$$S = 10.8\sin\theta + 0.03, \quad S < 9\% \tag{8a}$$

$$S = 16.8\sin\theta - 0.05, \quad S \geq 9\% \tag{8b}$$

where S is the inclined steepness factor and θ speaks to slant in degree. The A range for the topographic factor (LS) in the Sutlej catchment fluctuated from 0 to 19, as shown in Figure 7. Most of the investigation zone (95.27%) had the LS esteem extending between 10.1 and 18.9.

The slope length factor map was derived by applying Equation (5) to the flow accumulation map and slope length exponent (m) map in the raster calculator environment of GIS. Equation (5) was converted to a form of grid equation as given below:

$$L = \left(\text{Flow Accumulation} \times \text{Grid Size}/_{22.13}\right)^{m} \tag{9}$$

where grid size = 30 m, and slope length exponent m was taken from the m map for the respective grid.

### 3.3.4. Crop Management Factor (C)

The crop factor is the utmost essential constraint utilized in both USLE and RUSLE that demonstrates the impacts of vegetation and other land crops on the soil loss assessment [17,18]. The C factor can be defined as a fraction of the soil loss rate from a region with specified crop and management to that of an indistinguishable area [20]. In this study, LULC was prepared for the ISRO Geosphere Biosphere Program under the project entitled "Land Use/Land Crop Dynamics and Impact of Human Dimension in Indian River Basins" for the year 1995 on a scale of 1:2, 50,000 was used to derive the layers of various LULC parameters. The Sutlej basin has been classified into fourteen land use classes, namely, barren land, built-up land, crop land, deciduous broad leaf forest, evergreen broad leaf forest, evergreen needle leaf forest, grassland, mixed forest, permanent wetland, plantation, shrub land, snow and ice, waste land, and water bodies. Land use/land crop and their C factor values are shown in Figure 8, and statistics of each land use/land crop and their C and P factor values are presented in Table 4. The C values were used in the present study in concurrence with [30].

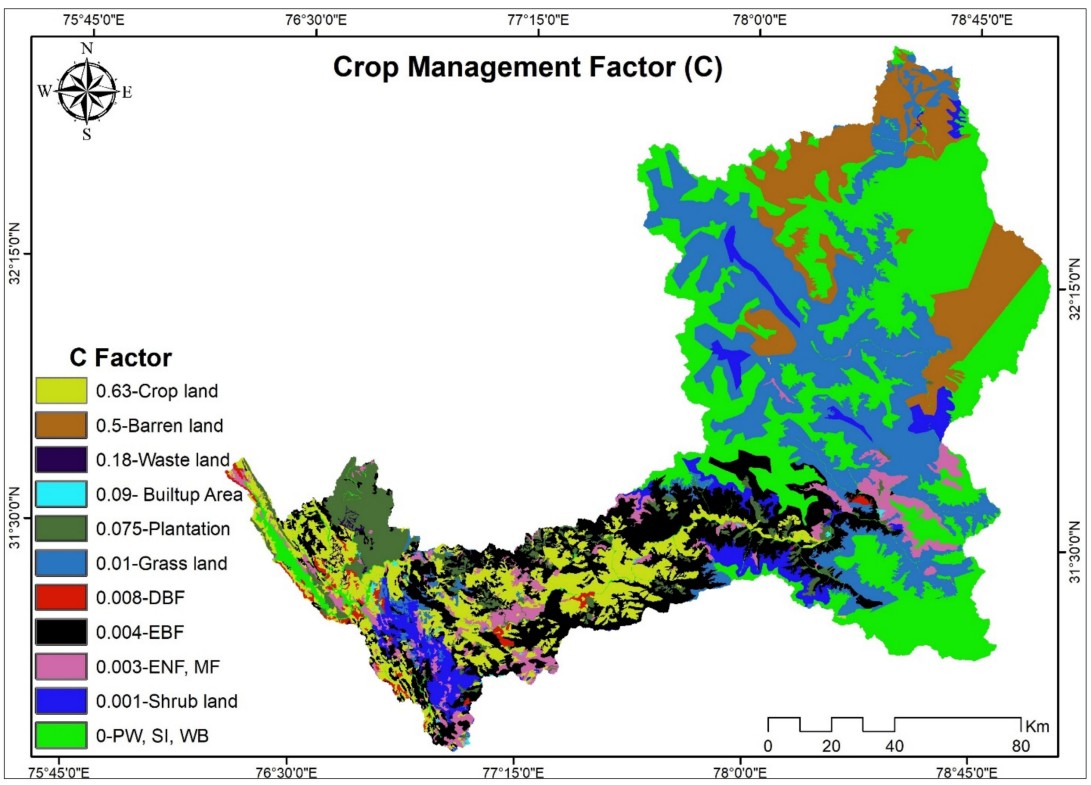

**Figure 8.** Crop management (C) factor with LULC.

**Table 4.** LULC statistics of the Sutlej catchment with C and P factors.

| S. No. | LULC Class | Area (ha) | Area (%) | C-Factor | P-Factor |
|---|---|---|---|---|---|
| 1 | Barren Land | 2244.23 | 10.40 | 0.5 | 1 |
| 2 | Built-up | 21.82 | 0.10 | 0.09 | 1 |
| 3 | Crop Land | 1872.21 | 8.67 | 0.63 | 0.28 |
| 4 | Deciduous Broad Leaf Forest | 158.63 | 0.73 | 0.008 | 1 |
| 5 | Evergreen Broad Leaf Forest | 2960.61 | 13.71 | 0.004 | 1 |
| 6 | Evergreen Needle Leaf Forest | 20.80 | 0.10 | 0.003 | 1 |
| 7 | Grassland | 4927.63 | 22.83 | 0.01 | 0.8 |
| 8 | Mixed Forest | 888.56 | 4.12 | 0.003 | 1 |
| 9 | Permanent Wetland | 1.25 | 0.01 | 0 | 1 |
| 10 | Plantation | 913.81 | 4.23 | 0.075 | 0.5 |

**Table 4.** *Cont.*

| S. No. | LULC Class | Area (ha) | Area (%) | C-Factor | P-Factor |
|--------|------------|-----------|----------|----------|----------|
| 11 | Shrub Land | 990.60 | 4.59 | 0.001 | 1 |
| 12 | Sown and Ice | 6276.08 | 29.07 | 0 | 1 |
| 13 | Waste Land | 52.80 | 0.24 | 0.18 | 1 |
| 14 | Water Bodies | 259.00 | 1.20 | 0 | 1 |

Classification accuracy of land cover is also included as per Table 4. LULC accuracy estimation based on 933 random sample was created for all 14 classes, and it showed an overall accuracy of 88.10%; error of omission and error of commission are also shown in Table 5. However, the kappa coefficient, originally developed to measure observer agreement for categorical data, was estimated to be 0.87, and this indicates good to excellent agreement of classified LULC classes.

**Table 5.** Accuracy assessment of the LULC map for the year 2005. (The changes in the various LULC classes are highlighted).

| Classification Data | BL | BU | CL | DBLE | EBLF | ENLF | GL | MF | PW | P | SL | S&I | WL | WB |
|---------------------|----|----|----|------|------|------|----|----|----|---|----|-----|----|----|
| BL | 67 | 0 | 0 | 0 | 0 | 0 | 3 | 0 | 0 | 0 | 6 | 0 | 0 | 0 |
| BU | 0 | 78 | 3 | 0 | 0 | 0 | 0 | 0 | 0 | 2 | 0 | 0 | 0 | 0 |
| CL | 0 | 1 | 108 | 1 | 0 | 0 | 0 | 1 | 2 | 0 | 0 | 3 | 0 | 2 |
| DBLE | 0 | 0 | 1 | 67 | 0 | 0 | 0 | 0 | 0 | 1 | 0 | 0 | 0 | 0 |
| EBLF | 4 | 0 | 0 | 1 | 48 | 0 | 0 | 0 | 3 | 0 | 0 | 0 | 2 | 0 |
| ENLF | 0 | 0 | 0 | 0 | 4 | 53 | 0 | 1 | 1 | 0 | 0 | 0 | 0 | 0 |
| GL | 0 | 0 | 0 | 0 | 0 | 0 | 34 | 4 | 0 | 0 | 3 | 0 | 0 | 0 |
| MF | 0 | 0 | 5 | 0 | 0 | 0 | 0 | 89 | 0 | 0 | 0 | 0 | 0 | 5 |
| PW | 3 | 5 | 0 | 0 | 0 | 0 | 0 | 0 | 27 | 0 | 1 | 3 | 0 | 0 |
| P | 0 | 0 | 0 | 0 | 0 | 1 | 6 | 0 | 0 | 53 | 0 | 0 | 0 | 0 |
| SL | 0 | 0 | 0 | 0 | 5 | 0 | 0 | 0 | 0 | 0 | 46 | 0 | 0 | 0 |
| S&I | 0 | 0 | 6 | 0 | 0 | 0 | 0 | 0 | 1 | 0 | 0 | 56 | 0 | 15 |
| WL | 2 | 0 | 0 | 0 | 0 | 0 | 0 | 0 | 1 | 0 | 0 | 0 | 37 | 0 |
| WB | 0 | 0 | 0 | 1 | 0 | 0 | 2 | 0 | 0 | 0 | 0 | 0 | 0 | 59 |

### 3.3.5. Conservation Practice Factor (P)

Practice factor (P) for RUSLE shows the surface condition that affects the flow pathways and flows hydraulics [22] and also reflects the erosion control estimates, for example, terracing and strip cropping. Estimation of practice factor, P, unlike management training, is given by [2,8]. Standard of P factor values range from 0.28 to 1, as shown in Figure 9, and assigned standards for each class are revealed in Table 3 for the Sutlej catchment, in which a higher value of P factor shows the minimum conservation practices, and a lower value of P factor shows good conservation practices.

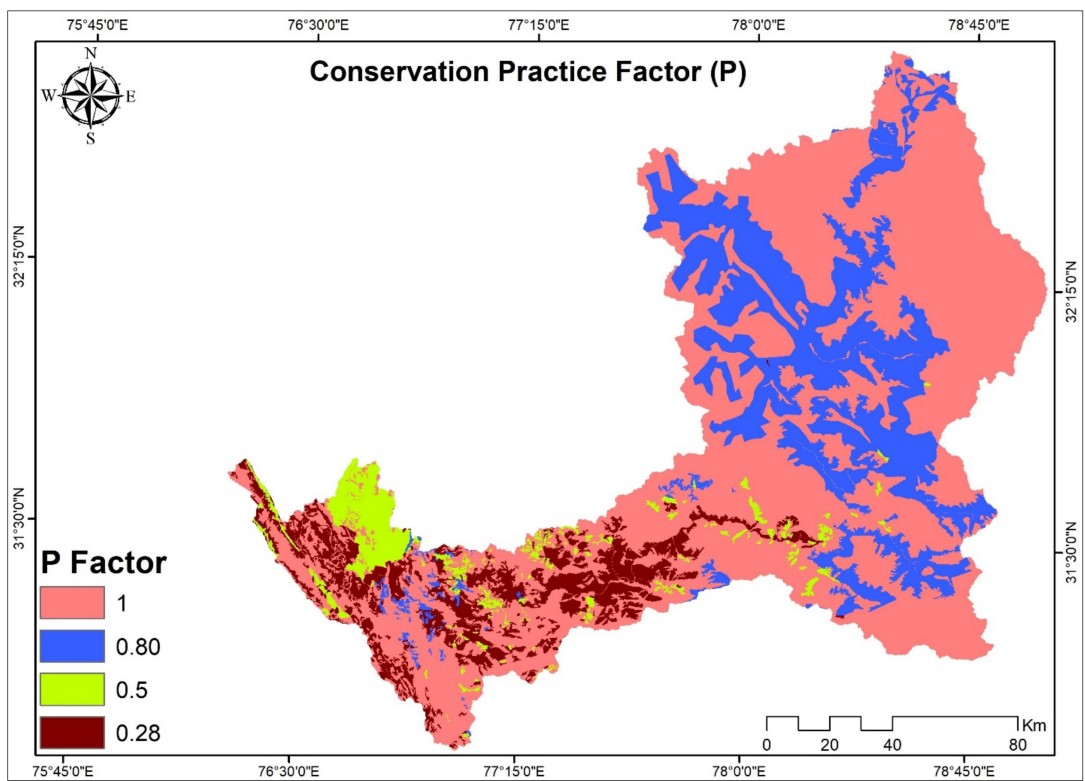

**Figure 9.** Spatial distribution of conservation practice (P) factor.

## 4. Results and Discussion

### 4.1. Suspended Sediment Concentration in the Sutlej Basin

This section represents the results of sediment yield in the form of erosion in the Sutlej catchment, which were obtained by analyzing rainfall (mm) and sediment concentration (gm/lit). The maximum suspended sediment concentration (mg/L) for the Sutlej River at Kasol, Suni, and Rampur occurred at Kasol on 1 August 2005 with 34,034.5, at Suni on 1 August 2000 with 18,362.4, and at Rampur on 1 August 2000 with 16,582.1, respectively. The dissemination of different size portions in the suspended sediment in the river system was in the order of fine > medium > coarse. The entire catchment is plagued with a serious problem of soil loss. Anthropogenic intervention, including construction of roads, mining, and other evolving ventures, had also accentuated the movement of soil erosion. Average discharge and suspended sediment concentration are shown in Figure 10 at Kasol for the period 1995–2005, at Suni for the period 1995–2005, and at Rampur for the period 1995–2003, respectively. Later on, validation was carried out for these three gauging sites for prevention measures towards erosion control in the prioritized areas of the Sutlej catchment.

### 4.2. Erosion Potential Map

For ranking the Sutlej sub-catchments and assessment of soil loss, layers were formed for R, K, LS, C, and P factors of the RUSLE model using the ArcGIS spatial analyst tool, specifically the raster calculator, to quantify, estimate, and create the maps of soil erosion and severity for the Sutlej catchment. These are the main components that govern soil erosion at a particular location in a catchment.

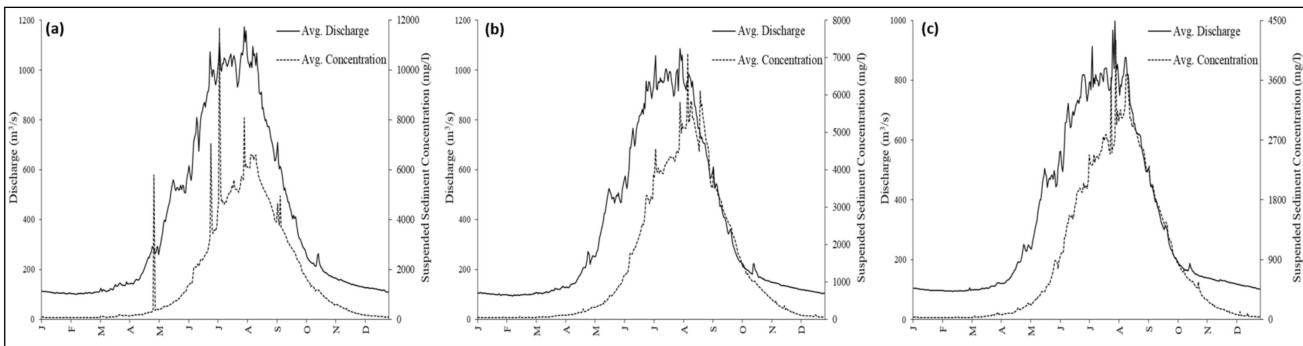

**Figure 10.** Average discharge and suspended sediment regimes in the Sutlej River at Kasol (**a**), Suni (**b**) FY (1995–2005), and Rampur (**c**) FY (1995–2003).

### 4.2.1. Variation of Factors (R, K, LS)

The estimated RE value ranged between 12.30–2131.98, 40.22–1144.36, and 427.07–2030.8 MJ mm/ha/h/year, with average values of 467.29, 326.43, and 554.28 MJ mm/ha/h/year for the years 1995, 2005, and 2015, respectively. This study area was classified into six textural classes, namely glaciers, glaciers and rock outcrops, loamy, loamy skeletal, rock outcrops, and sandy. The estimated K factor values for the diverse textural classes were 0, 0.01, 0.27, 0.19, 0.02, and 0.44 t-h-ha/ha/MJ/mm, respectively. Ref. [48] mentioned the K factor values for the lower mountainous catchment of the lower Himalayas, ranging from 0.09 to 0.48 Mg-h/MJ/mm. The topographical (LS) measure values for the Sutlej catchment varied from 0 to 18.9. The LS factor imitates the impact of incline length and slant steepness on the collapse rate in the catchment in view of an adjustment in slant and length attributes.

### 4.2.2. Optimization of Crop Management Factor (C) and LULC Classes

In order to characterize the crop management values (C), the Sutlej catchment was classified into seven as well as 14 land Use/land Crop classes using ISRO-GBP data. The crop management value for the northern Himalayan region varied from 0 to 0.63. The closer agreement between observed and simulated average annual soil loss was obtained using C as 0.63 and 14 land use classes. The erosion map, as shown in Figure 11, was created using varying C values from 0.2 to 0.63 and seven and 14 LULC classes. It can be seen from the figure that the erosion computation was highly sensitive to the number of LULC classes and adequate choice of C. Therefore, it is essential that calibration must be exercised to identify the appropriate number of classes along with the optimum C.

### 4.2.3. Variation of Conservation Practice Factor (P)

Using the LULC and the practice factor, four classes of conservation factor were documented. The P factor varied from 0.28 to 1 for this catchment; a higher value of P factor showed the minimum conservation practices, and a lower value of P factor showed good conservation practice. Applying the RUSLE model, a higher value of average annual soil loss (A) indicated a higher potential of soil erosion in the cell, which was specified as 30 × 30 m for this study. The average annual soil is presented in Table 6, which has been noticed in the Sutlej catchment for the period of 1995 to 2015 up to the Bhakra Dam. This RUSLE model was validated at that point by comparing the simulated yield with the observed yield for the period of 1995 to 2005 at Kasol and Suni and 1995 to 2003 for Rampur, respectively, as shown in Figure 12. It is worth mentioning that higher erosivity was associated with higher precipitation amounts, as shown in Figure 13.

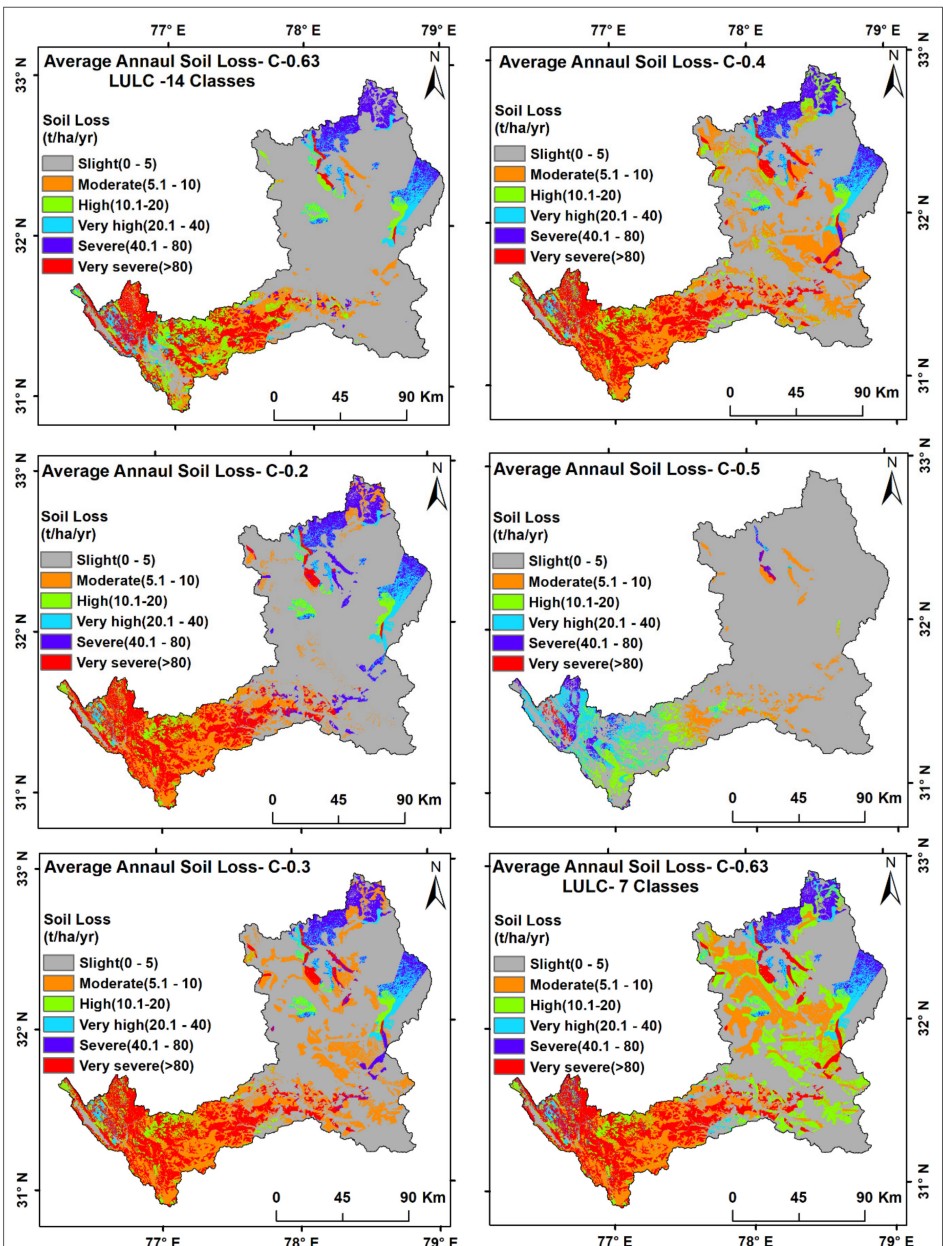

**Figure 11.** Dependence of average annual soil loss with LULC classes and crop factor C.

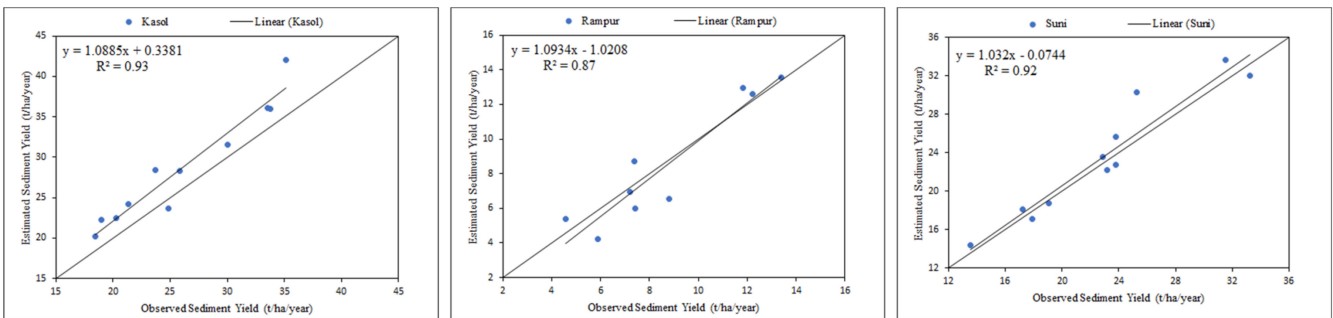

**Figure 12.** Observed and simulated sediment yield at Kasol, Rampur, and Suni.

**Table 6.** Average soil loss in the Sutlej catchment up to the Bhakra Dam FY (1995–2015).

| Year | Average Soil Loss (t/ha/Year) |
| --- | --- |
| 1995 | 79.67 |
| 1996 | 48.14 |
| 1997 | 54.93 |
| 1998 | 49.42 |
| 1999 | 52.68 |
| 2000 | 52.85 |
| 2001 | 42.34 |
| 2002 | 36.16 |
| 2003 | 55.95 |
| 2004 | 36.37 |
| 2005 | 55.71 |
| 2006 | 45.55 |
| 2007 | 64.84 |
| 2008 | 74.11 |
| 2009 | 46.08 |
| 2010 | 62.44 |
| 2011 | 75.55 |
| 2012 | 72.64 |
| 2013 | 61.78 |
| 2014 | 42.54 |
| 2015 | 70.02 |

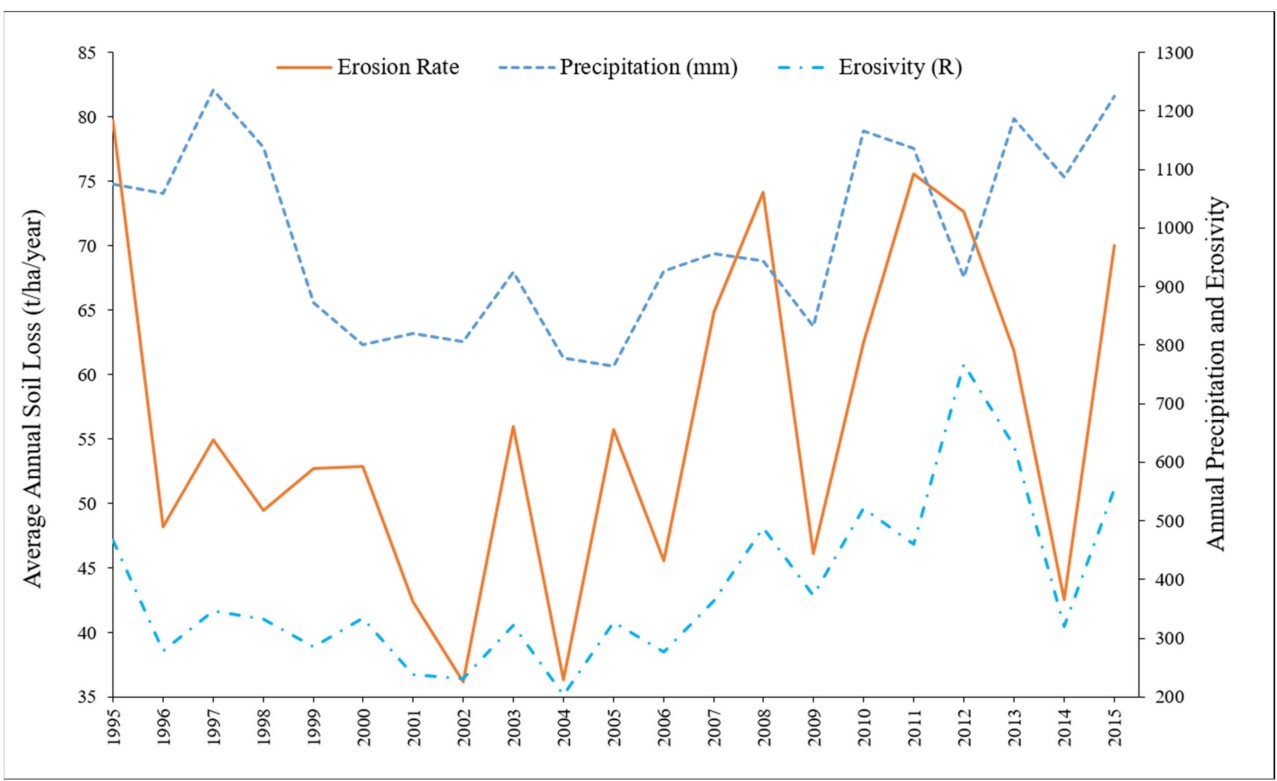

**Figure 13.** Erosion rate and its relationship with precipitation and erosivity FY (1995–2015).

Figure 13 shows the link between erosion rate and precipitation in the Sutlej catchment. However, Figures 14 and 15 relate to the soil loss and severity zones in the catchment. The GIS and RS technologies were helpful in the present investigation to quantify the soil loss from the catchment and in the prioritization of sub-watersheds in the Sutlej catchment.

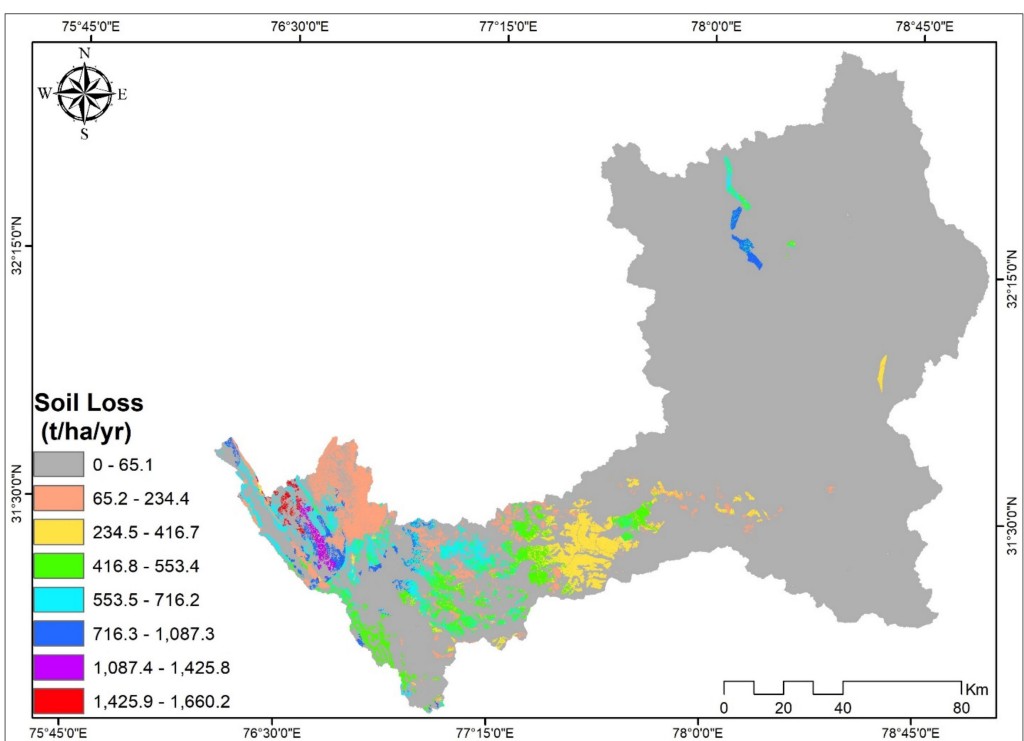

**Figure 14.** Spatial distribution of soil loss in the catchment (C = 0.63 and 14 LULC classes).

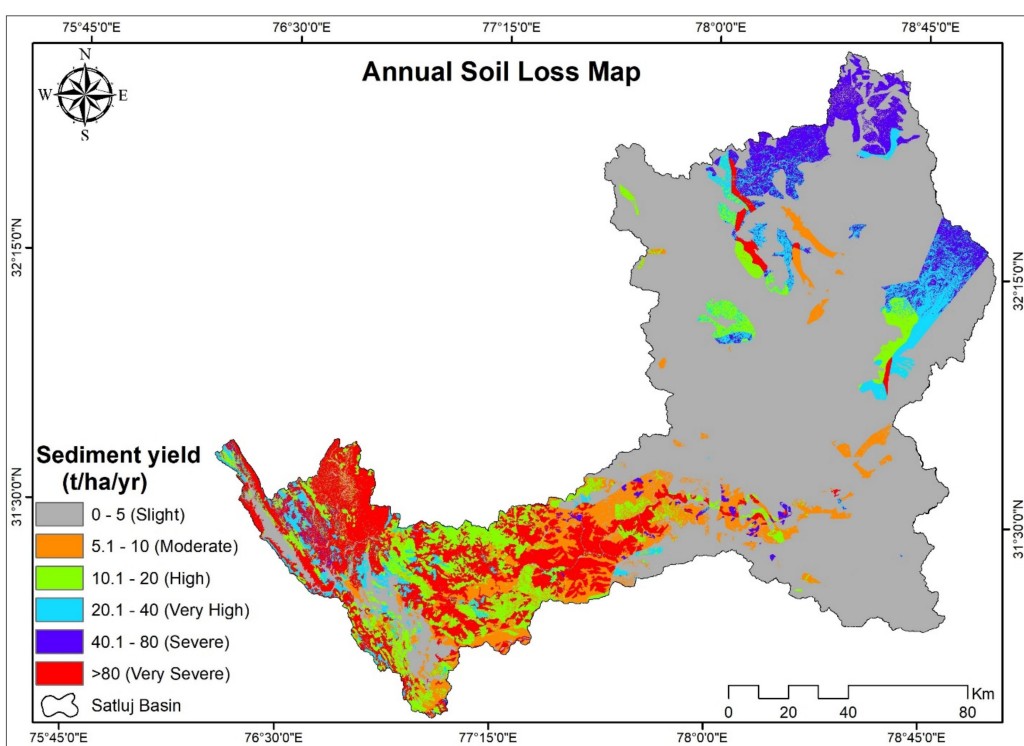

**Figure 15.** Distribution of average annual soil loss in the catchment (C-0.63 and 14-LULC classes).

*4.3. Land Use/Land Crop Change Effect on Soil Erosion*

For analyzing the effect of land use/land crop on soil loss in the catchment, an attempt was also made to reclassify LULC in seven classes, as shown in Table 6. The crop management values for the increased agricultural land, forest, and water bodies in the catchment were derived from remote sensing images, as shown in Table 7. The

increment in agricultural land showed a remarkable consequence on the erosion rate in the Sutlej catchment. It was observed that the erosion rate was not affected by forest area and wet land in the catchment, because the C-factor did not change much for forest and waterbody or wet land. After an increase in the agriculture land, the average soil erosion increased to 90.57 t/ha/year compared to 56.18 t/ha/year. The difference was about 34.39 t/ha/year, which was likely due to the larger increment in agriculture land, and the other two categories, i.e., forest area and wet land, did not show much effect on the annual average soil loss in the catchment up to the Bhakra Dam. In comparison to the actual annual average soil loss, there was a percentage increase of 37.97 t/ha/year when compared with the actual soil loss. After reclassification of satellite images, agriculture crops covered 40.30% of the total catchment area, which had been 8.67% as a crop land with the C-factor having a value of 0.63. However, with seven classes, the agreement between observed and computed sediment loads was not better than it was with 14 classes.

**Table 7.** Land use/land cover classes and corresponding C-factor for reclassified data.

| S. No. | LULC Class | Area (ha) | Area (%) | C-Factor | P-Factor |
|--------|------------|-----------|----------|----------|----------|
| 1 | Agriculture | 8699.81 | 40.30 | 0.63 | 0.28 |
| 2 | Barren Land | 2246.87 | 10.41 | 0.5 | 1 |
| 3 | Built-up | 21.81 | 0.10 | 0.09 | 1 |
| 4 | Forest | 4023.95 | 18.64 | 0.003 | 1 |
| 5 | Snow and Ice | 6282.35 | 29.10 | 0 | 1 |
| 6 | Waste Land | 52.63 | 0.24 | 0.18 | 1 |
| 7 | Water Bodies | 259.58 | 1.20 | 0 | 1 |

*4.4. Identification of Priority Areas in the Sutlej Basin and Erosion Probability Zones*

After verifying Horton's laws for stream area, stream length, and stream numbers, nine sub-watersheds created in the entire Sutlej catchment and the erosion-affected areas were generated by the weighted index method in GIS using the spatial analyst tool and prioritization of sub-catchments defined into six classes, which was recommended by [2,8] for Indian conditions, as follows: slight, moderate, high, very high, severe, and very severe, in units of t ha$^{-1}$ year$^{-1}$. It was observed that about 62.91% of the area of the catchment was falling below the minor erosion class (Table 8). The regions exposed to moderate, high, very high, serious, and very serious erosion impending classes were 7.42%, 7.75%, 5.36%, 5.14%, and 11.42% of the whole topographical area, respectively.

**Table 8.** Area of soil erosion classes in the Sutlej catchment.

| S. No. | Rate or Erosion (t/ha/Year) | Area (ha) | Area (%) | Class of Priority |
|--------|------------------------------|-----------|----------|-------------------|
| 1 | 0–5 | 13,580.12 | 62.91 | Slight |
| 2 | 5–10 | 1601.17 | 7.42 | Moderate |
| 3 | 10–20 | 1672.00 | 7.75 | High |
| 4 | 20–40 | 1157.47 | 5.36 | Very high |
| 5 | 40–80 | 1109.78 | 5.14 | Severe |
| 6 | >80 | 2465.75 | 11.42 | Very Severe |

Thus, from a degradation point of view, the final four categories of soil erosion in the river basin need immediate attention. According to the average yearly soil loss from the sub-catchments, the SW9, SW7, SW8, SW6, SW2, SW1, SW4, SW2, and SW3 should be prioritized in that order. Primacy levels indicate how we should care for the watershed area with plants and conservation efforts. Table 9 and Figure 16 offer a prioritization of Sutlej catchment sub-watersheds. The current literature also includes references to physics-based research [1] that makes use of sediment transport and deposition. This article does not fully explore the possibilities of such methods; future work will determine whether physics-based models provide any advantages over the more standard RUSLE approach.

**Table 9.** Prioritization of sub-catchments.

| S. No. | Sub-Watershed | Soil Erosion Class | Rank |
|--------|---------------|--------------------|------|
| 1 | SW1 | Very high | 6 |
| 2 | SW2 | Very high | 5 |
| 3 | SW3 | Slight | 9 |
| 4 | SW4 | Moderate | 7 |
| 5 | SW5 | Slight | 8 |
| 6 | SW6 | Severe | 4 |
| 7 | SW7 | Very severe | 2 |
| 8 | SW8 | Very severe | 3 |
| 9 | SW9 | Very severe | 1 |

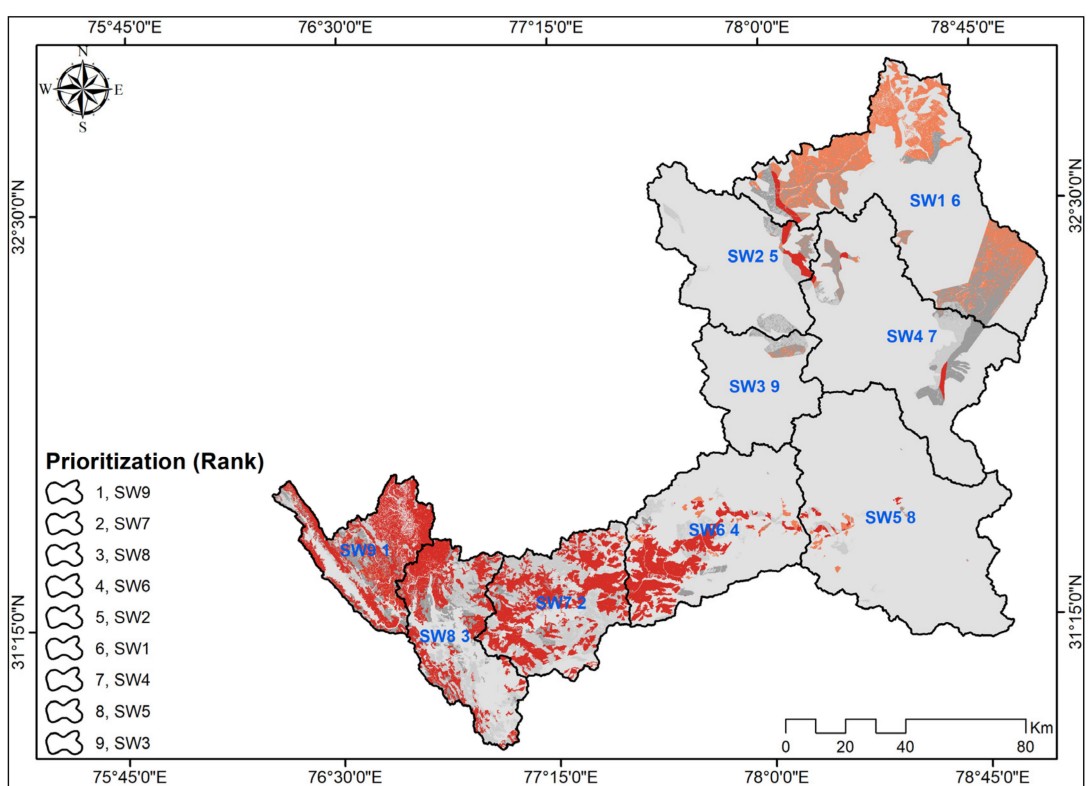

**Figure 16.** Prioritization of sub-watersheds in the Sutlej catchment (C-0.63 & 14-LULC classes).

On top of that, researchers have shown that the lower Sutlej watershed loses an average of 56.18 tons of soil per hectare each year due to erosion and soil loss. About 29.67% of the catchment area falls into the high to extremely severe categories of soil erosion, according to the visualization. Thus, this analysis shows prioritization of the catchment to be treated in that manner.

## 5. Conclusions

A quantitative appraisal of normal yearly soil loss for the Sutlej catchment was made on a pixel-by-pixel premise with a GIS-based implicit RUSLE equation considering precipitation, soil, land use, and topographic datasets to identify areas susceptible to serious erosion in the catchment for management planning purposes. The average annual soil loss from the Sutlej catchment was found to be 56.18 t/ha/year. The soil erosion map shows that about 29.67% of the catchment area falls into the high to very severe categories. Sensitivity analysis by varying crop factor values also indicated an optimum value of 0.63 for 14 LULC classes. In the Sutlej as a whole, three gauging sites showed the best agreement between observed and predicted sediment output after using this optimized crop component. The

data given here also show a direct correlation between the quantity of precipitation and the sediment production that has been recorded.

**Author Contributions:** Conceptualization, S.G.; methodology, S.G. and C.S.P.O.; software, S.G.; validation, S.G., C.S.P.O. and V.P.S.; formal analysis, S.G.; investigation, S.G.; resources, S.G. and C.S.P.O.; data curation, S.G.; writing—original draft preparation, S.G.; writing—review and editing, S.G., C.S.P.O., V.P.S., A.J.A. and S.K.J.; visualization, S.G., C.S.P.O. and V.P.S.; supervision, C.S.P.O.; funding acquisition, C.S.P.O. All authors have read and agreed to the published version of the manuscript.

**Funding:** This research received no external funding.

**Institutional Review Board Statement:** Not applicable.

**Informed Consent Statement:** Not applicable.

**Data Availability Statement:** Data is contained within the article.

**Conflicts of Interest:** The authors declare no conflict of interest.

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
