# Peer review of "Pixel-Based Soil Loss Estimation and Prioritization of North-Western Himalayan Catchment Based on Revised Universal Soil Loss Equation (RUSLE)"

_sustainability, doi:10.3390/su152015177_

Round 1

Reviewer 1 Report

1.   Introduction:Break down this lengthy text into smaller paragraphs, each focusing on a specific theme or viewpoint. This will enhance readability and comprehension.For the hydrogeology of the Sutlej basin, you provided detailed information about the South Asian monsoon and snowmelt. However, it might be necessary to explain more clearly how the South Asian monsoon affects the hydrological characteristics of the basin, and how snowmelt impacts river runoff. This will help readers who are not familiar with the region better understand.In the introduction, clearly state the research's objectives, background, and significance. Emphasize how the study addresses knowledge gaps and why it's important for understanding soil erosion and land use changes in mountainous regions.While discussing soil types and erodibility, you might need to elaborate more on how different soil types impact erodibility. Consider providing more details on the sensitivity and effects of different soil types on erosion.

2.   Study Area and Datasets:The section begins with a clear description of the study area and its geographical features. However, consider adding a sentence or two to connect this information with the broader context of the study and how it relates to soil erosion and land use changes.

3.   Methodology:When describing the climate of the Sutlej basin, you mentioned different climate zones. However, it might be helpful to explain in more detail how these different climate types influence soil erosion and water flow.It is recommended to include a concluding paragraph summarizing the above-mentioned content at the end of this methodology section.

4.   Results:The discussion of the results appears to be overly generalized, with each subsection presented separately. It is recommended to synthesize the different factors and interrelate their implications.The descriptions of information within the result charts and graphs are limited. It is advisable to provide more comprehensive explanations for each visual aid before proceeding with the analysis.Figure 12, adjustments to the color palette and line thickness should be considered to enhance clarity and comprehensibility.Figure 13, the visual representation of content seems unclear. It is suggested to utilize distinct colors for different scenarios and emphasize critical nodes on the graph.Figure 16 exhibits information that appears cluttered. It is advised to modify font color or size to improve legibility. Table 4, enhancing the significance of the Accuracy metric can be achieved through the use of a heatmap or varying colors to highlight crucial information.

Reviewer 2 Report

Analysis of the effect of land use changes on erosion rate and sediment yield is particularly useful to identify critical areas and define catchment-area treatment plans. This study utilized remote sensing and GIS techniques combined with the Revised Universal Soil Loss Equation (RUSLE) on a pixel basis to estimate soil loss over space and time, and prioritized areas for action. The research was complete and the figures were beautiful overall. However, the manuscript was in a rather messy formation now, the authors must appropriately deal with the following major and minor comments.

1). What is the initial innovation of the article? In current version, the manuscript just applied the RUSLE for the soil erosion evaluation. Did the focus was Pixel-based? There should have certain innovations in Methodology for the Sustainability Journal with high citations.

2). The Introduction part, now it seems a report in current form. The authors should point out the key methodology and innovation of this study.

3). Maybe it need a Discussion part in the manuscript?

4). It has a lot of writing errors in the manuscript, for example, the GIS presented firstly in the Abstract should been given the full spelling. Lines 53, 54, the USLE and RUSLE. Lines 65, 66, Land Use/Land Cover.

5). Figure 5, why did the map present the rainfall distribution for the four classes, we know your focus was the data formation in Pixel level, and would you give a figure with continuous change in space? Also the R factor.

6). Figure 10, what’s your reason for the presentation of the discharge and suspended sediment regimes only for 2005 and the average value of 1995-2005. And the sub-figure (c) was the year 2000?

Reviewer 3 Report

Dear Authors

It is an interesting work. This research is significance to study the the spatio-temporal variations of soil erosion and the relevant factors in an area affected by an earthqueake. The work is acceptably presented, with sound methodological approach, sin embargo no se incluyen unos resultados well-supported (i.e. con referencias). I suggest revising the manuscript (i.e., major revision) focusing mainly in the results and discussions.

Round 2

Reviewer 2 Report

   Although the quality of the manuscript has been greatly improved after the revisions, there are still some issues to be improved. 

   (1) the 5 Summary and Conclusion. Why do not the Conclusions?

   (2) the overall format of the text. It need to be hierarchical of the interpretation of the paragraph. Why not insert the picture into the text following the Journal‘s guidence.?

Author Response

I've made the changes accordingly.

Reviewer 3 Report

It is a very interesting work. This research is of great significance to study soil loss estimation in a Himalayan catchment based on Revised Universal Soil Loss Equation. The work has improved considerably after the first review. Now, the work is nicely presented, with sound methodological approach, well-supported results and conclusions

Author Response

I've made the changes accordingly.
